# A Blockchain-Enabled Secure Digital Twin Framework for Early Botnet Detection in IIoT Environment

**DOI:** 10.3390/s22166133

**Published:** 2022-08-16

**Authors:** Mikail Mohammed Salim, Alowonou Kowovi Comivi, Tojimurotov Nurbek, Heejae Park, Jong Hyuk Park

**Affiliations:** Department of Computer Science and Engineering, Seoul National University of Science and Technology (SeoulTech), Seoul 01811, Korea

**Keywords:** digital twin, blockchain, smart contracts, botnet detection, cybersecurity

## Abstract

Resource constraints in the Industrial Internet of Things (IIoT) result in brute-force attacks, transforming them into a botnet to launch Distributed Denial of Service Attacks. The delayed detection of botnet formation presents challenges in controlling the spread of malicious scripts in other devices and increases the probability of a high-volume cyberattack. In this paper, we propose a secure Blockchain-enabled Digital Framework for the early detection of Bot formation in a Smart Factory environment. A Digital Twin (DT) is designed for a group of devices on the edge layer to collect device data and inspect packet headers using Deep Learning for connections with external unique IP addresses with open connections. Data are synchronized between the DT and a Packet Auditor (PA) for detecting corrupt device data transmission. Smart Contracts authenticate the DT and PA, ensuring malicious nodes do not participate in data synchronization. Botnet spread is prevented using DT certificate revocation. A comparative analysis of the proposed framework with existing studies demonstrates that the synchronization of data between the DT and PA ensures data integrity for the Botnet detection model training. Data privacy is maintained by inspecting only Packet headers, thereby not requiring the decryption of encrypted data.

## 1. Introduction

The Industrial Internet of Things is modernizing Smart Cities with its capacity to assist in the rapid decision-making process of optimizing the manufacturing methods and facilitating the flow of goods. The management of raw materials, manufacture of goods, assembly-line production, final packaging and warehousing of goods are interconnected using heterogeneous devices [1,2]. Smart Factories are enriched by the automation of the production cycle delivering reduced times to produce goods, increase business profits, and promote further opportunities in innovation for greater flexibility and improved efficiency [3,4,5]. Device-to-Device communications further enable a comprehensive overview of the industrial environment where sensors collaborate with each other, sharing data for the further optimized performance of industrial machines. Decision-making processes are further expedited by reducing the time to compute data from Cloud to Fog/Edge environments [6,7,8].

The Industrial Revolution (IR) from IR 3.0 to IR 4.0 supported by IIoT has greatly fueled the growth of Smart Cities with a rapid response to the growing demands of its economy [9,10,11]. However, the transition to Industry 4.0 has increased the risks to network security due to weak and ineffective security protocols in IIoT devices. The reduced computation capacity and battery power restrict the implementation of strong security methods directly on the devices [12,13,14]. Moreover, the heterogeneity of devices implemented in a single Smart Factory restricts a common security protocol implementation for the system-wide network. Botnet attacks present a dual risk to the Industry 4.0 network [15,16]. Firstly, the compromise of devices using brute-force attack methods risk exposure of private data to unauthorized entities. Furthermore, the security of data is compromised, as cyber attackers modify and inject corrupt data, affecting the network’s performance. Secondly, Distributed Denial of Attacks (DDoS) further risks the entire network’s operational performance, resulting in failure device data computation, data modeling, and analysis for operational performance [17,18].

Cloud-based Intrusion Detections Systems using Artificial Intelligence (AI) detect and identify anomalies in the IIoT traffic, but they are generally ineffective for time-sensitive applications such as Manufacturing Factories that rely on high efficiency to outmatch consumer demands [19]. DT technology has recently been explored for the early detection of malicious behavior in the network traffic originating from the physical IIoT devices. DTs are virtual twins of physical devices and synchronize data amongst each other using bi-directional communication channel. Modifications made to the physical devices are mirrored in their DTs and vice versa [20,21]. Furthermore, their integration with AI models encourages advanced modeling techniques for the early detection of malicious behavior such as botnet activity before devices are fully weaponized to launch full-scale DDoS attacks on the network servers.

In this paper, we propose a Blockchain-enabled Digital Twin Framework for the early detection of botnet activity in an IIoT environment. The Framework integrates DT, physical IIoT devices, Deep Learning model, Blockchain, and Smart Contracts for securing the data flow of a Smart Factory environment. The framework implements a private blockchain managed by a security vendor to register DTs, and a virtual node, PA, which is responsible for securely synchronizing the DT data with physical devices using Smart Contracts. Network traffic monitoring using Deep Learning inspects and analyzes both encrypted and unencrypted traffic using packet headers. The model inspects for the early detection of botnet behavior, alerting the security vendor to modify network policies and isolate other Digital Twins from the infected group of devices.

The motivation for writing this paper is to address the growing concerns of botnet-based cyberattacks on the Industry 4.0 network. The main contributions of this paper include the following.

A DT environment of the IIoT enables implementation of robust security protocols that leverage resources from the edge datacenters.A private blockchain registers DTs as transactions in Blocks preventing malicious nodes from injecting corrupt data into the data stream, affecting the data integrity of the collected data.A virtual node, PA, is registered with the Private Blockchain and captures the network using a boundary set by the security vendor to detect Replay attacks on DTs.DT synchronization is established securely with the PA to identify if any packets between the physical IoT and the virtual twin are dropped. The protocol detects man-in-the-middle attackers between physical–virtual twins.The Deep Learning botnet-detection model inspects unique IP addresses and half-open connections, indicating an open communication channel between the device and the botnet Command and Control server.A comparative analysis with existing research shows that the proposed framework provides data security, integrity, privacy, device availability, and non-repudiation.

The remainder of this paper is organized as follows. Section 2 discusses the existing state-of-the-art studies in securing Industrial networks against cyberattacks. In Section 3, we present the overview of the proposed system and discuss its workflow process in detail. In Section 4, we perform a comparative analysis of the proposed scheme with recent studies. The weaknesses of other schemes and future research areas are discussed, and, finally, in Section 5, we conclude our paper.

## 2. Related Work

Botnet detection has been extensively researched in recent years; however, they do not fulfill the growing requirements of the growing IIoT environment. In this section, we discuss the various state-of-the-art related works and present the key areas of consideration for a secure and reliable framework designed for the early detection of botnet activity.

### 2.1. Existing Research

Several botnet-detection studies for IoT devices focus on implementing an AI model using both centralized and distributed classification methods. A Zero-day botnet-detection method for IoT–Edge devices is presented by Popoola et al. [22], implementing a Federated Learning approach for traffic-anomaly detection. Several local models are trained directly on IoT–Edge devices using the Deep Neural Network model to ensure data privacy concerns. After several rounds of training on local devices, the global model is updated using the aggregated results of local models. A Bot–IoT dataset is used to simulate a botnet scenario achieving high detection rates but requires a long training time compared with other centralized training models. A two-fold machine learning approach by Hussain et al. [23] focused on identifying botnet behavior prior to a cyberattack against the network. A dataset generated from three publicly available datasets includes 33 scan types and 60 DDoS attacks. Using a dual machine learning approach, the first ResNet-18 model scans local IoT devices for scanning activity indicative of botnet behavior. The second ResNet-19 model focuses on identifying DDoS attacks and alert the system of a botnet-based attack. Though the research indicates that early botnet behavior detection is essential for network security, its dual machine learning approach still relies on identifying a DDoS attack for a botnet detection behavior. An automated approach for botnet detection in IoT devices is presented by Trajanovski [24] integrates honeypots with a sandbox environment. The objective is to analyze devices with varying software and hardware configurations and prevent attackers by performing anti-forensics and anti-analysis techniques. Identified attacking devices are blacklisted from connecting with the network. Several Command-and-Control servers use botnets as broadcast devices for a cyberattack, which may include critical devices for a Healthcare, Manufacturing, or an Internet of Vehicles environment. The blacklisting of devices is unsuitable for modern Smart City environments. A visualized botnet detection framework proposed by Vinayakumar et al. [25] focuses on identifying botnets in a Smart City environment. A two-level deep learning model first identifies the most frequent DNS queries performed on Ethernet connections in conjunction with a threshold value. A Deep-Learning-based Domain generation algorithm classifies benign and malicious domain addresses.

Blockchain-based anomaly-detection solutions in IoT environments focus on securing the network using distributed and scalable solutions. Hayat et al. [26] implemented a Machine-Learning-based DDoS attack-mitigation method by initially registering all devices in the decentralized network. The study suggests ejecting malicious devices failing to authenticate with the Blockchain network. Each interacting device requires prior registration with the network before inter-communicating with other devices. The proposed approach does not address that several botnets are pre-registered IoT devices with the Blockchain network; thus, an attacker is already an existing network member. The study by Lekssays et al. [27] suggests that modern botnets implement a P2P architecture to increase their chance of infecting a wider area of IoT devices. The authors proposed a dynamic botnet-detection method using Blockchain for establishing trust between IoT devices and internet service providers. The privacy of various actors is preserved by associating pseudo-IDs for each member. The detection model requires a consensus among all blockchain members to eject the device from the network upon the detection of malicious activity. Sun et al. [28] proposed a lightweight permissioned Blockchain network for IoT devices where each device is assigned a public key based on its identity attributes to filter authorized devices from invalid requests. To avoid the delay in data processing in the decentralized environment, a policy decision point algorithm identifies Blockchains’ node-making real-time off-chain decisions. Xu et al. [29] proposed a Bi-directional attack proof Blockchain using chameleon hash functions for the IoT environment. The Committee Members Auction consensus algorithm provides high scalability and attack resistance of the decentralized network. Secret sharing distributes secret keys to network members via Smart Contracts, ensuring only authorized members participate in the network.

### 2.2. Comparison Study

In this sub-section, we describe the open challenges of the existing research and compare them with the contributions of the proposed Blockchain-enabled Secure Digital Twin framework. We examine how the proposed framework further improves the data security of the IIoT environment with Deep Learning models

First, we discuss and compare the contributions of studies that focus on botnet detection by implementing AI- and Honeypot-based systems with the proposed framework. The Federated-Learning-based framework proposed in [22] relies on training data as local models on the IoT devices, preventing user data transfer on edge networks for data privacy. The objective of this study depends on retaining the raw IoT data on the device layer, preventing external entities from accessing user data. IoT devices with low computational resources are incapable of running resource-taxing deep learning models; thus, the feasibility of the study remains unanswered. Furthermore, the research does not evaluate the performance impact of directly implementing computationally expensive deep learning models directly on resource constraint IoT devices. The two-fold machine learning approach in [23] initiates DDoS attack detection after the IoT devices are compromised. The study’s objective is focused on increasing the DDoS’s attack-detection accuracy and does not address the security or privacy of IoT devices and data. The IoT botnet-detection framework presented in [24] does not inspect IoT devices for growing malicious behavior and instead relies on a honeypot environment for active botnet behavior detection. The study’s objective is to avoid the attacker’s detection of a Honeypot environment and not to prevent its existence and growth. The Visualized botnet-detection system proposed in [25] relies on training the deep learning model using DNS data on a centralized server. The transmission of IoT data from IoT devices presents data security, integrity, and privacy risks. A man-in-the-middle attack easily compromises the deep learning model using poisoning attacks.

The proposed framework implements the botnet-detection deep learning model on the edge layer, utilizing the vast available computational resources compared to the device layer. Training on the edge layer reduces the risk of the poor performance of the deep learning model training on IoT devices as present in [22]. Even though the proposed framework implements a centralized deep learning approach, the PA-DT synchronization reduces the risks of poisoning attacks affecting the botnet-detection system. Implementing the DT synchronization with the PA reduces the risks of poisoning attacks present in the studies [23,25]. The synchronization protocol of DT-PA prevents the compromise in the deep learning training dataset and maintains the detection accuracy of the botnet training model. Furthermore, the synchronization protocol identifies any man-in-the-middle attacks, thus enabling the early detection of cyberattacks on the IIoT environment. In the proposed framework, DT certificates enable a botnet-spread-prevention policy in the IIoT environment managed by the security vendor. Upon identification of an infected DT, the security vendor revokes its corresponding IoT devices from communicating with other DTs. The honeypot-based study in [24] relies on botnet behavior detection. Still, it does not present a method to contain the spread of the malicious scripts transmitted from compromised IoT devices to benign devices. A security vendor is forced to either let the IoT devices function without any security policy or shut down all devices until a security firmware is patched, thus impacting operational performance in the IIoT environment.

Finally, the remaining research focuses on deploying blockchain technology as the primary security model for the secure authentication of IoT devices and secure storage of IoT data. The Blockchain-based Multilevel DDoS Mitigation mechanism proposed in [26] requires each device to possess a signature for communicating with the server and avoid being blocked from communicating. Signature verification of each device using Smart Contracts with Blockchain introduces delays due to frequent transaction requests in a real-time scenario. Thus, the model is unsuitable for IIoT environments such as Smart Factories, Healthcare, and Smart Energy-based environments. The PAutoBotCatcher framework presented in [27] registers devices in the blockchain following whitelist and blacklist mechanisms. The framework successfully blacklists all infected devices using their IP addresses; however, a man-in-middle-attack using a spoofed IP address is able to transmit flooding attacks on the server. As the new IP address is blacklisted, the attacker dynamically changes their address and resumes the attack. The Blockchain-based IoT access system presented in [28] proposes a localized, decentralized ledger for establishing which IoT devices have access to offload data for computation at the edge layer. The research focuses on preventing unauthorized entities from accessing the system server. However, the lack of an active traffic flow detection protocol prevents the system from identifying the growing botnet activity, resulting in registered and authenticated devices launching DDoS attacks. The Bidirectional-linked Blockchain system presented in [29] prevents cyber attackers from taking control of the blockchain network using secret keys. The study’s main objective is to present a robust defense for the system server but does not secure the IoT devices directly. A committee-member-based consensus mechanism selects which users are selected to join the network. A lack of security measures between the edge layer and the device-layer-compromised devices enables malicious users to transmit corrupt data to the system server.

In the proposed framework, IoT devices do not individually connect with the server but do so using a DT, where each DT is allotted an authentication certificate. A single DT enables a collective cluster of associated IoT devices from communicating with the network. The DT-based certificate approach addresses the challenge present in [26] of frequent authentications, resulting in network congestion due to poor transactions/minute in the Blockchain system. The time-boundary-based DT-PT synchronization protocol actively identifies compromised IoT devices by inspecting transmitted packets. The vulnerability of an attacker using a spoofed address present in [27] is identified early by the security vendor before a large-scale cyberattack takes place. The study in [28] establishes access control for IoT devices using blockchain but lacks a robust traffic-anomaly detection-protocol. Addressing this vulnerability, the proposed framework implements a dual traffic-anomaly-detection process, the DT-PA synchronization protocol and the deep-learning-based botnet-detection model. The dual-anomaly-detection approach actively identifies man-in-the-middle attacks such as Delay-and-Replay attacks and detects DDoS attacks using Deep Learning. Traffic flows to the network. The active DT-PA synchronization protocol identifies registered IoT devices in a DT transmitting their attack packets and limits the growth of botnet behavior by revoking the DT certificate, thus addressing the vulnerability found in [29], where registered devices continue to transmit attack packets.

### 2.3. Key Considerations

A botnet-detection framework for IIoT devices in Smart Factories requires a secure data synchronization method and the quick identification of malicious activity in the network. The following five key considerations are essential to our proposed framework and their comparative analysis is illustrated in Table 1.

Data Security—Data transmission from IoT devices to virtual twins at the edge layer require authorization measures preventing malicious entities from accessing confidential Industrial data. A framework based on sharing data ownership to untrusted entities requires an authentication mechanism to ensure only trusted nodes have access.Data Integrity—Deep Learning models require accurate captured data to train and detect botnet open channels in the IoT devices. A botnet-detection framework requires a data verification protocol to verify if data in transmission are modified. Modification of data at the virtual twins through man-in-the-middle attacks reduces the reliability in the accurate detection of True positives and True negatives.Data Privacy—Industrial data secrets shared via IP packets risk being exposed when analyzed using Deep Learning methods. A privacy-preserving framework focusing on Packet Header information reduce the risk of data exposure to unauthorized entities.Availability—Devices identified with botnet activity risk infecting other sensors using brute-force methods. Successful detection of infected devices should be isolated from spreading botnet scripts to other benign devices. The availability of benign devices is essential not to halt the entire industrial network operation.Non-Repudiation—Malicious packet data shared by infected devices are required to be securely stored in tamper-proof environments such as Blockchain. Infected devices ID and IP address are essential for record-keeping for future security firmware updates.

## 3. Proposed Framework

This section presents the overview and the components used in the Digital Twin enabled DDoS attack-prevention framework for IoT enabled Smart Cities. The device layer consists of several physical entities such as Factories, Hospitals, and Vehicles that operate independently supported by embedded sensors. A DT of each entity is maintained at the edge layer to monitor the network flow. A security vendor is responsible for maintaining the security of the DT and operates a Deep Learning model to analyze the DT behavior for anomalous behavior. Attack-detection at the DT improves the scalability of the network and reduces attack-detection time. The fine-grain inspection of each device’s traffic increases the computational load over time, such as when a hospital or factory adds the number of sensor devices in its network based on its demand.

A Consortium Blockchain network maintained by the security vendor is responsible for recording the identified malicious activity indicative of growing Botnet activity on the Blockchain. As the botnet targets are not necessarily on the same cloud server of the infected entity, the security vendor is required to submit the alert to other security vendors and a Cyber-Crime unit of the developing botnet activity. Furthermore, in the workflow sub-section, we illustrate and present the detailed process flow of the botnet-detection process, DT ranking, Blockchain data storage, and the Smart Contract initiated to alert other security vendors. Table 2 illustrates the abbreviations used in the framework.

### 3.1. Overview of the Proposed Framework

The proposed scheme consists of the several elements participating in the botnet-detection process, DT ranking and Profile generation, and network alert of botnet-generation process.

IoT devices are identified as physical devices that gather and upload data to the cloud for processing. Each device is responsible for generating data from varying sources such as machinery, transport vehicles, assembly lines, and private devices in homes and offices. Devices are heterogeneous with varying security protocols, computing power, and battery power.Using the example of a Smart Factory, different manufacturing processes, such as raw material management, assembly line production, and packaging, deploy several sensors. Each process deploys several sensors that communicate data with other processes. The DT of each separate process, such as assembly-line production, is designed on the edge layer that synchronizes and mirrors data flowing in the local network. A DT provides a surveillance for all devices and access points within its cluster. Data collected at this layer trains the local Deep Learning model to actively search for botnet-spread behavior. Computation power is provided by local edge-based datacenters to the DT, thus reducing the necessity of running Deep Learning models on IoT devices with low computational power and battery capacity.A Security Vendor monitors the DT using the Deep Learning model and maintains the ranking profile for each DT under their supervision. In this paper, we assume the Security Vendor is responsible for maintaining and monitoring the network security of four different DTs of a Smart Factory environment.Security Vendors manage a private blockchain for storing the rank of each DT identified with a high risk of botnet activity. Each security vendor operates separately and monitors several organizations, such as a group of Hospitals or Factories.Individual security vendors initiate Smart Contracts for alerting other security managers of a possible security attack. Records are submitted of each contract as a transaction in the Blockchain to ensure the chain of custody.

As illustrated in Figure 1 and presented in Algorithm 1, the proposed framework is based on the following process.

Step 1.IoT devices in the factories collect data from the machinery and share with the edge for computation and analysis.Step 2.DTs present at the edge layer synchronize data with their respective departments in a factory, such as the production floor ($DT_{1}$), raw-materials management ($DT_{2}$), assembly line ($DT_{3}$), and packaging and warehousing ($DT_{4}$). Each department is represented by an application which collects raw data from IoT devices and forwards to the edge for processing. For example, the production floor collects all data from the sensors deployed in the manufacturing and processing of goods.Step 3.Ensuring only authorized DTs are allowed to send data, they are registered in the private blockchain. A separate PA is registered, which facilitates data-synchronization between the DT and IoT devices.Step 4.DT and the PA synchronize data to ensure industry traffic is not intercepted by unauthorized entities. Data collected by the DT are used for training the model for botnet detection and requires security from man-in-the-middle attacks.Step 5.DTs represent the network topology, traffic load, and the benign and malicious traffic flowing in their network. Data are captured at this junction to analyze and identify botnet activity.Step 6.TCP/UDP packet headers are collected to analyze unencrypted packet data. IP packets such as Hypertext Transfer Protocol Secure (HTTPS) are Secure Socket Layer (SSL) encrypted and thus the Botnet detection identified using packet headers inspects both HTTP and HTTPS traffic.Step 7.Features are collected from the DTs that include both TCP and UDP packets. Packet time to arrive, open or closed connections, and the IP addresses used are collected for analysis.Step 8.In this study, to simulate a botnet detection, malicious packet flow is introduced in $DT_{1}$. $DT_{2}$, $DT_{3}$, and $DT_{4}$ are benign virtual cluster of devices and are potential targets of growing botnet activity.Step 9.Successful classification of $DT_{1}$ with botnet activity, the security vendor initiates prevention measures for the growing botnet activity. All traffic from the infected device DT is isolated from inter-communicating with devices from the benign DTs and prevents the botnet activity’s spread. As each DT includes all devices and local gateway access points such as routers, all outbound traffic at $DT_{1}$ is excluded from sending packets to $DT_{2}$, $DT_{3}$, and $DT_{4}$.Step 10.Each Smart Contract is recorded in the Blockchain to maintain the chain of custody during investigations into the source of attacks and device security firmware management.

**Algorithm 1** The proposed Blockchain-enabled Secure Digital Twin Framework**Input:**Device1−n, DT1−4, PA, TCP/UDP packets**Initialization Phase:**Distribute devices (Device1−p, Device1−q, Device1−r) based on factory processes.Assign factory process 1 – 4 to DT1−4Split Devices to DT1, DT2, DT3, and DT4.Register DTn (Devicesn) in the private blockchain and assign certificate
DT1−n transmits and shares data with other DT1−nSynchronize data between DT1−4 and PA to determine       
if (DTpackets= PApackets)              return network secure         **else**              return register IP addresses and DTn in private blockchain         **end if**Collect TCP/UDP packet headersTrain deep learning model using collected packetsAssign IoT devices their group based on their process         **if** botnet activity = false              return network secure         **else**              return register IP addresses and infected DTn in private blockchain              revoke compromised DTn certificate to prevent inter *DT* communication         **end if**  **End**

IoT devices include varying types of embedded sensors with differing battery and computational resources. A common DT ensures security protocols are feasible across all devices despite device heterogeneity. IP headers provide data from both from encrypted and unencrypted packets. The Blockchain network prevents malicious nodes from sending false data to the network, raising multiple false alarms preventing security vendors from learning true attack scenarios.

### 3.2. Workflow of the Proposed Scheme

This section describes the process flow of the proposed scheme based on four phases. First, DT and a PA are registered with the Blockchain to ensure only authorized virtual twins receive data from IoT devices. Secondly, an authorized PA ensures the secure synchronization and capture of IoT packet data for DT synchronization. Thirdly, traffic inspection of packets using Deep Learning are analyzed for botnet activity. Lastly, DT isolates inter-DT communication to prevent the growth of the botnet traffic and initiates the collection of infected data using a certificate generated by the PA for each DT.

#### 3.2.1. Packet Auditor and Digital Twin Registration

Each PA and DT are first registered with the Blockchain network to prevent malicious nodes from transmitting false alarms to the security vendor. We assume the PA is a secure virtual node and monitored by the Security Vendor. As illustrated in Figure 2, PA and DT registration process flow is as follows.

Step 1.The security vendor generates a new and unique Packet Auditor ID ($PA_{ID}$) using a nonce value for randomness and the first five values of the hash value generated based on the number of devices paired with the DT.Step 2.A transaction ($Txn_{1}$) is created in the Blockchain network using the new $PA_{ID}$. The $PA_{ID}$ is first encrypted using a public key generated ($P_{PubK}$).
(1)$$Txn_{1}=P_{PubK}\left(PA_{ID}\right)$$Step 3.The private blockchain node verifies if the network has an existing PA registered. If a $PA_{ID}$ exists, the transaction will not proceed.Step 4.If $PA_{ID}$ is not found, then the new $PA_{ID}$ is registered using a Smart Contract. A new block in the network is created, and, as such, other blocks in the Blockchain are aware of it.Step 5.Once registered a certificate (${\mathrm{Cert}}_{\mathrm{DT}}$) is generated for each DT associated with the PA, a second transaction is created ($Txn_{2}$) to register the new certificate using the Private Key ($P_{PrvK}$) belonging to the PA. A certificate ensures that all future intercommunicating DTs can validate if the other DT is not a malicious virtual twin and not part of the network.
(2)$$Txn_{2}=DT_{pubK}\;\left(P_{PrvK}\left(PA_{ID}\right)\right)$$Step 6.Once $Txn_{2}$ is processed using the public key of the DT ($DT_{pubK}$), $DT_{1}$ obtains the certificate using its private key ($DT_{prvK}$). Step 7.The certificate is distributed to $DT_{1}$. Each new certificate using the $P_{PrvK}$ of PA generates $Txn_{n}$ for $DT_{n}$.Step 8.The public keys of the DTs are stored in the Blockchain to prevent a malicious node from spoofing as a valid DT and attempting to access or corrupt private data.Step 9.The PA is further responsible for recording the IP addresses of each IoT device included in $DT_{n}$ and registering them on the Blockchain. This step prevents an attacker from joining their malicious node as part of a valid $DT_{n}$. 

The secure registration of each DT and the PA enables the security vendor to establish that data transmission in the DT are from valid nodes. The recording of IP addresses prevents an external entity from adding nodes into the DT and transmitting corrupt data. Furthermore, the secure and immutable records stored in the private blockchain prevents an external entity from interacting with other IoT devices and injecting scripts.

#### 3.2.2. Digital Twin Synchronization

As both PA and DT are registered in the Blockchain in the network, the synchronization between DT and IoT devices is monitored. The objective of the synchronization monitoring is to periodically verify if data in transmission between the DT and its devices are not modified in transmission. Any modification is indicative that private data have been intercepted and monitored by unauthorized entities. The process to synchronize data between the DT and PA is based on two phases. In the first phase, the PA captures the data lifecycle of the devices within a DT. Secondly, the DT shares its packet flow captured at a defined state and requests the PA to synchronize and validate the data.

As shown in Algorithm 2, the process flow for the first phase, capturing data life cycle is based on the flowing steps.

Step 1.The PA generates a new profile for each DT included in its network. Each DT in a factory undergoes various phases of the product-development lifecycle, and a vast amount of data is generated. Profiles stored in the DT are kept on a temporary phase and not stored on the Blockchain. Once a DT synchronizes the data with the PA, the profiles are removed. However, in the event of an attack detection, the PA captures profiles and actively stores them in the private blockchain and establish records of malicious activity for future investigation.Step 2.The PA is supported by cloud resources; however, capturing real-time data on the PA incurs a very high storage cost and thus a boundary is established which is followed by both the PA and DT during synchronization.Step 3.For each DT, a new profile ($Prof_{DT}$) is generated, where the timestamp of captured packets ($tstmp_{i}$), their Device IDs ($D_{ID}$), and the IP addresses of packet sources ($IPsrc_{i}$) and destinations ($IPdst_{i}$) are collected as part of an individual Profile. Here, i represents the captured packets.
(3)$$Prof_{DT}=D_{ID}\left(IPsrc_{n},\;IPdst_{n},\;tstmp_{n}\right)$$The $Prof_{DT}$ is a collection of the basic information of each packet collected. The packet payload is not collected and stored due to storage concerns and operational costs. Step 4The data collection time boundary is selected by the security vendor to reduce the amount of data collected. Each boundary limit varies in days, hours, and minutes, where packet data are collected. Each time-boundary consists of an upper ($bd_{upper}$) and lower boundary ($bd_{lower}$) to ensure a precise data collection period and storage limitations.
(4)$$DT_{n}=(bd_{upper},\;bd_{lower}\;\left(Prof_{DT}\right)$$

Each boundary limit set by the security vendor by the PA is also applicable on the DT for accurate synchronization. The security vendor is responsible for initiating the synchronization process, after which all ProfDT in the PA are removed.
**Algorithm 2** Digital Twin synchronization**Input:**ProfDT,  DTn, DID, IPsrcn, IPdstn, tstmpn, bdupper, bdlower, PA, CertDT, DTprvK, DTpubK, PAID, PPrvK, PPubK**Begin:**Design new ProfDT**:**Select time boundaryVendor = bdupperhh:mm:ss, bdlowerhh:mm:ssAssign time boundary to ProfDT   for (DT=1 to DTn) do
   ProfDT (bdupper, bdlower) = Vendor (bdupper, bdlower)    **End for****Authenticate:**Encrypt DT CertificateEncrypt CertDT with DTprvKDTn(DTprvK(CertDT)) → BlockchainDecrypt and Verify DT Certificate   if
 DTn(DTpubK(CertDT)) = True
    return DT authentication successful   else    CertDT not found. Authentication failed   **end if**Encrypt PA IDEncrypt PAID with PPrvKPA(
PPrvK)) → BlockchainDecrypt and Verify PA ID   if PA (PPubK (PAID) = True          return PA authentication successful; initiate DT Sync   else        PAID not found. Authentication failed   **end if****Initate Sync**Verify DTn ProfDT with the PA    if
 ProfDT (bdupper, bdlower) = Vendor (bdupper, bdlower)         return Initiate Sync      if
 PADIDIPsrcn, IPdstn, tstmpn = DT DIDIPsrcn, IPdstn, tstmpn            return packets match          DT DIDIPsrcn, IPdstn, tstmpn → Blockchain         else               Profiles mismatch. Update security vendor.             DT DIDIPsrcn, IPdstn, tstmpn)) → Blockchain           **end if**        else            return ProfDT profile mismatch. Update security vendor        **end if****End**


The PA temporarily stores the data until the DT successfully synchronizes its collected data. In the second phase, the DT synchronization process is initiated. The process flow for synchronizing the packet data between the DT with the PA, as illustrated in Figure 3, are as follows.

Step 1.Prior to the synchronization process, DT initiates a sync request by transmitting its $Cert_{DT}$ signed by its $DT_{prvK}$. We assume entities in the network operate in an untrusted environment, where each node is suspected to be a malicious node.Step 2.The private blockchain network verifies the $Cert_{DT}$ using the stored $DT_{pubK}$. If the certificate is found valid, a Smart Contract is initiated between the DT and the PA. If, however, the $DT_{pubK}$ fails to verify the certificate, it is assumed to be an invalid request.Step 3.The PA submits its $PA_{ID}$ signed using its $P_{PrvK}$. The Blockchain network verifies the $PA_{ID}$ using the stored $P_{PubK}$. If the $PA_{ID}$ is found valid, the Smart Contract’s security conditions are fulfilled, and the synchronization process initiates.Step 4.The $Prof_{DT}$ stored in the DT and the $Prof_{DT}$ stored in the PA share the same time limit of captured data, and, thus, the DT is required to only share the $DT_{n}$ along with the time boundaries, $bd_{upper},\;bd_{lower}$, that are the start and end timestamps, when packets are recorded.Step 5.The PA analyzes the $DT_{n}$ to locate the correct profile and verifies if the $bd_{upper},\;bd_{lower}$ matches with its stored $DT_{n}$. If it matches, the PA transmits the $DT_{n}$ with the DT and fulfills its Smart Contract condition.Step 6.The DT based on each $D_{ID}$ data flowing in its network matches the Device IDs shared by the PA, such that,
(5)$$PA\left(D_{ID}\left(IPsrc_{n},\;IPdst_{n},\;tstmp_{n}\right)\right)=DT\;\left(D_{ID}\left(IPsrc_{n},\;IPdst_{n},\;tstmp_{n}\right)\right).$$Step 7.If the $bd_{upper},\;bd_{lower}$ does not match in the $DT_{n}$, then an invalid request is sent as a reply to the DT, the Smart Contract is closed. If the boundaries match, data synchronization proceeds normally.Step 8.Packet flow between the DT and the PA is synchronized, and data are verified.Step 9.The results of the contract are registered in the Blockchain as a transaction, and the outcome of the synchronization is recorded.

If the packet data matched, the synchronization-state protocol is successful, and the process will repeat itself. If, however, the timestamps varied due to differing time boundaries, the framework records a malicious activity in the DT. The timestamp between the packets received by the PA from the IoT device and those recorded by the DT indicates a Delay attack.

#### 3.2.3. Network Traffic Monitoring

The packet Inspection of both TCP and UDP packets include features that are both unencrypted compared to their payloads and do not require deep inspections of payloads to reduce the time to train the Deep Learning (DL) model. Feature extractions are identified to include packets from IP addresses that are unique, i.e., devices transmitting data to unknown sources that are indicative of two outcomes, the attack target server is external to the factory’s cloud server, or the devices are communicating with external IoT nodes to inject malicious scripts. Bots transmit malicious scripts to different IP addresses that are not part of the local network and target vulnerable sensors belonging to other networks, such as Healthcare, Logistics, and Office Buildings. The framework focuses on inspecting random IP addresses for inspection, which are expected to exhibit higher malware traffic than legitimate traffic flow. Furthermore, the number of random IP addresses is higher, as bots do not transmit scripts to the same IP address repeatedly to avoid detection and to infect several devices in the shortest time and initiate a DdoS attack as soon as possible.

Other features selected for inspection include the monitoring of the half-open connections between devices as bots open several connections with other devices with invalid requests without receiving a reply to ACK messages. Such methods indicate a possible TCP SYN flood attack, where an attacker attempts to make a server unavailable by flooding it with several requests, leaving all ports of the target in half-open state. Furthermore, we include the maximum, minimum, and average size of packets in our feature classification, as several malwares transmit fewer headers compared to benign packets. Smaller packets require more packets to be transmitted, resulting in a flood-based attack on the target server. The Maximum Transmission Packet size is set at 1500 bytes, whereas attacks packets send packets of 90 bytes [30], increasing the count of the packets transmitted. Lastly, the time difference in packet transmissions is inspected, as reduced time intervals indicate an increased count of packets transmitted to the destination server compared with legitimate traffic.

Botnets maintain an open connection of TCP traffic with the Command and Control (CnC) server by exchanging PUSH and ACK messages, a behavior exhibited by the Mirai botnet actively sending commands and tracking the count of devices included in the botnet [31]. The detection of infected devices is identified, demonstrating frequent connections with unique IP addresses, indicative of two scenarios, a connection with an attack target or a connection with the C&C server. We correlate the results of devices that have frequent half-open connections with unique IP addresses as the source of the C&C server.

In this paper, we implement the Long-Short-Term Memory (LSTM), a variant of the Recurrent Neural Network ideal for a large collection of packet data and where the data captured have high similarity. Since the amount of traffic originating from IoT devices is considered large in size, we implement the LSTM model for the botnet training. The advantage of LSTM over other models is where they maintain a memory cell that preserves the earlier sequential input data and manage long-term dependencies. The model consisting of input, forget, and output gates collects long-term dependencies and uses a sigmoid function to filter data at each gate.

#### 3.2.4. Device Isolation

In the final phase of the framework, infected devices are recognized by their IP addresses. The Security Vendor is alerted to the presence of a botnet and issues a network policy update for device isolation. The process flow for device isolation, as presented in Algorithm 3, is as follows.

Step 1.The IP addresses of infected devices are recorded and analyzed with the records stored in the Blockchain along with their associated DT. The certificate of the DT is used to isolate the device from communicating with other DTs.Step 2.The Security Vendor initiates a network-wide policy update by initiating a Smart Contract with DTs ($DT_{2},\;DT_{3},$ and $DT_{4}$). Each DT is required to prove their validity by sharing their certificates, which have been signed using their private keys $DT_{prvK}$.Step 3.The Blockchain network verifies the certificate of each DT using their respective $DT_{pubK}$. Failure to verify the validity of the Certificate results in the connection being terminated.Step 4.Once the proof of identity is validated, the security vendor issues a revoke command of $DT_{1}$’s certificate, preventing any inter-device communication to prevent the spread of botnet scripts.Step 5.The revocation of the $DT_{1}$’s certificate is stored in the Blockchain along with the policy update issued to other DTs.

**Algorithm 3** Device Isolation**Input:**$Prof_{DT},\;\;DT_{n},\;{\mathrm{Cert}}_{\mathrm{DT}}$, $DT_{prvK}$, $DT_{pubK}$, Security Vendor
**Begin:**$\mathrm{Revoke}\;DT_{n}$:
$\;\;\;\;\mathrm{for}\;(\mathrm{DT}=1\;\mathrm{to}\;DT_{n}$) do       Encrypt DT Certificate$\;\;\;\;\;\;\mathrm{Encrypt}\;{\mathrm{Cert}}_{\mathrm{DT}}$ with $\;\;\;\;\;\;DT_{prvK}$$\;\;\;\;\;\;DT_{n}(DT_{prvK}({\mathrm{Cert}}_{\mathrm{DT}}$)) → Blockchain    Decrypt and Verify DT Certificate$\;\;\;\;\;\;\;\;\;\;\;\mathrm{if}\;DT_{n}(DT_{pubK}({\mathrm{Cert}}_{\mathrm{DT}}$)) = True
               return DT authentication successful$\;\;\;\;\;\;\;\;\;\;\;\;\;\;\;\;\;\;\mathrm{if}\;DT_{n}(({\mathrm{Cert}}_{\mathrm{DT}}$)) = True             // Verify if $DT_{n}$ is compromised$\;\;\;\;\;\;\;\;\;\;\;\;\;\;\;\;\;\;\;\;\;\;\mathrm{return}\;\mathrm{revoke}\;DT_{n}(({\mathrm{Cert}}_{\mathrm{DT}}$))
$\;\;\;\;\;\;\;\;\;\;\;\;\;\;\;\;\;\;\;\;\;\;DT_{n}(({\mathrm{Cert}}_{\mathrm{DT}}$)) → Blockchain // Record revoked Certificate in Blockchain
                 else$\;\;\;\;\;\;\;\;\;\;\;\;\;\;\;\;\;\;\;\;\;\;\mathrm{return}\;DT_{n}(({\mathrm{Cert}}_{\mathrm{DT}}$)) is verified                 **end if**         else$\;\;\;\;\;\;\;\;\;\;\;\;\mathrm{return}\;{\mathrm{Cert}}_{\mathrm{DT}}$ not found. Authentication failed         **end if**     **end for****End**

The revocation of the DT’s certificate mirrors the policy on the physical devices, and, as such, all physical devices associated with $DT_{1}$ are blocked from communicating with $DT_{2},\;DT_{3},$ and $DT_{4}$. The secure recording of the actions taken by the framework on the private blockchain network ensures the maintenance of the chain of custody of all steps taken by the security vendor to secure the factory environment from future cyberattacks.

## 4. Analysis

In this section, we evaluate the performance of the proposed Blockchain-enabled Secure Digital Twin Framework for early botnet-behavior-detection with existing studies. We first analyze the performance of the PA- and DT-synchronization protocol for early detection of man-in-the-middle attacks. The performance is evaluated based on latency and accuracy of the synchronization protocol and compared with a baseline model. Secondly, we study the impact of an ongoing cyberattack on the transaction processing speed for increased validation of PA and DT and compare it with the existing studies. Finally, we analyze the impact of a DdoS attack on the Blockchain nodes affecting the transaction processing speed. We analyze the attack resistance on the PoA algorithm used in the proposed framework’s private blockchain and compare it with existing studies. Finally, using the feature selection based on the traffic identified from the PA-DT synchronization, we compare the botnet-detection model’s accuracy, precision, recall, and F1 scores with existing studies.

### 4.1. Evaluation Environment

The proposed framework is evaluated using a system running an Ubuntu 18.0.4 operating system using a 4.0 Ghz i7 processor and 64 GB RAM. Smart Contracts are designed using Solidity for authenticating DT and PA. The Private Blockchain is designed using the Proof of Authority consensus model where the block validator is limited to only the security vendor. We removed the requirement for the mining of blocks using complex cryptographic puzzles and enabled the security vendor to be the sole authority to approve new block mining. Physical IoT devices are represented using 5 Raspberry PI B+ with 2 GB RAM with 1.5 Ghz processor. A sixth Raspberry device represents the smart factory manufacturing process application, receiving traffic from other IoT devices. Using VirtualBox, we make a DT of the 6th Raspberry Device, the Physical Twin (PT) of which represents the DT on the edge layer. Data is mirrored between each PT IoT device and its virtual counterpart, DT, for maintaining synchronization. The PA uses a Docker Container with Wireshark to monitor collected packet data.

The Botnet-detection protocol is evaluated using the Bot–IoT Dataset [32,33,34,35,36,37] generated by the Cyber Range Lab of UNSW Canberra. The traffic dataset contains a total of 72 million records consisting of both benign and malicious botnet traffic. Attack types are focused on the most frequent of cyberattacks, DDoS and DoS attacks. The dataset is based on an 80:20 split based on training and testing sets, respectively. As stated in Section 3.2.3 of Network Traffic Monitoring, we focus on the count of half-open TCP connections and the maximum and the minimum counts of packets based on the uniqueness of their IP address. We include these connections on the premise that they are more likely to connect with external servers and launch cyberattacks. Furthermore, the feature selection also includes the mean count of packets for each unique IP address. Other features include the mean packet length to identify if the attacker attempts to launch a flooding attack using a small packet size. The feature selection is based on identifying the relevant features and excluding redundant data that serves no benefit in the botnet detection. Overfitting is a challenge, and the LSTM model implements a dropout value of 0.2 and uses two hidden layers. The count of hidden nodes increases from 23 to 64 and finally to 128. The evaluation of the model is based on 100 rounds to demonstrate its performance using the identified features.

### 4.2. Digital Twin and Packet Auditor Analysis

We first measure the impact on performance during the DT synchronization with the PT, which helps us evaluate the real-time application of DTs for other time-sensitive applications such as Healthcare. Second, we observe the latency and CPU consumption in packet synchronization between the DT and PA and compare it with a baseline model that detects packet exchanges in real-time. In this stage, we introduce malicious packets in the data stream of DT to evaluate the duration of identifying transmission of malicious information by the PA.

As illustrated in Figure 4a, the delay in terms of the recording of the packet stream on the DT is minimal compared to the real-time traffic in PT. An average delay of 0.002 milliseconds (ms) enables the DT to mirror real-time operations for time-sensitive tasks. Table 3 outlines the performance of the PA-DT synchronization protocol.

From Figure 4b, we observe that the CPU utilization of the PT and DT during state synchronization is nearly the same. Both DT and PT are set to use the same amount of CPU resources as the 1.5 Ghz processor based on the Raspberry PI B + device. Table 4 illustrates the analysis of the DT-PT synchronization CPU consumption.

We set the $Prof_{DT}$ using Equation (4) to analyze the captured data by the PA and compare with the data stream captured by the DT. As both DT and PA are separate entities, PA is designed using cloud resources using a Container with Wireshark operating. The application captures data and checks to verify if Equation (5) is true. For the sake of the experiment, we set eight different $Prof_{DT}$, i.e., the duration of the upper boundary is set at 0 days, 0 h, 10 min, and 0 s. The lower boundary is set at 0 days, 0 h, 1 min, and 0 s. The remaining $Prof_{DT}$ values are set consecutively till 0 days, 0 h, 2 min, and 0 s, and the lower boundary remains unchanged. Malicious traffic is injected in the DT data stream, increasing the flow of the packet to cause a flooding attack.

As shown in Figure 5a, we observe that, the larger the selected upper boundary, the higher the time difference in analyzing the collected packets from the DT increases. To offset the performance impact, the security vendor has the ability to modify both the upper and lower boundary limits as stated in Equation (3),
$$DT_{n}=(bd_{upper},\;bd_{lower}\;\left(Prof_{DT}\right).\;$$

We compared the performance of the PA with a baseline model using real-time packet analysis between the PA and the DT. The detection time between the baseline model and the proposed framework are nearly identical, where the real-time model outperforms packet analysis by 0.003 s. Table 5 outlines the performance of all eight $Prof_{DT}$, where the lowest difference recorded is 0.002 s, and the highest time difference is 0.005 s. We attribute the higher performance of the baseline model due to the active detection of variation in packets. In Figure 5b, we observe a high consumption of CPU resources in the baseline model, where the average difference is 6.73% between the models. The proposed framework outperforms the baseline model due to the lower and upper boundary limits that define when the PA packet analysis process is initialized.

From the evaluation of the Latency and CPU consumption of both the DT-PT mirror transmission and the PA analysis of the packet inspection with the DT, we observe that the proposed framework consumes fewer network resources in terms of CPU utilization and has low latency in maintaining a virtual profile of the Smart Factory PT on the edge layer for real-time operations. Furthermore, the proposed framework identifies malicious scripts or poison attacks during DT-PT transmission, thus ensuring data security and integrity.

### 4.3. Blockchain Scalability and Attack Resistance

In this section, we observe the private blockchain’s performance in terms of scalability in managing the growing count of the authentication requests of PA inspection by the security vendor. Secondly, we analyze the transaction speed and the attack resistance of the consensus model during active DDoS attacks.

The private blockchain in the proposed framework deploys the Proof of Authority consensus model, where the Security Vendor is the sole validator of the transactions and blocks. As the process is automated, it does not require manual approval from the validator to approve each to be block mined and transaction processed. However, the validator does have the manual override option to prevent the formation of new locks in the event of a DDoS attack or a 51% attack on the network. From Figure 6, we analyze the transaction frequency when the Smart Factory is under a cyberattack and the Security Vendor requires frequent recording of authentication approvals for recording new PA-DT packet audit analysis. The PoA maintains a steady average transaction speed of 20 s to process and record each request of PA and DT authentication requests in the block. We attribute the improvement in performance of the PoA over existing studies that implement the Practical Byzantine Fault Tolerance [27,28] and the Committee Members Auction [29] consensus models, due to the restriction of using only a single Validator. Furthermore, we kept the block size at 300 kb enabling the storage of more transactions per block and reducing the requirement to mine more blocks. Smaller block sizes require higher transactions per second. As the number of transaction requests increases to 200, the time consumed to approve transactions remains steady at 20 s. The Practical Byzantine Fault Tolerance implemented by Lekssays et al. [27] and Sun et al. [28] require 440 s. The Committee Members Auction implemented by Xu et al. [29] performed better than other models due to its 2 s mandatory waiting time; however, it performed poorly compared to PoA due to its higher number of validators.

The Blockchain network under a cyberattack such as a DDoS attack or 51% attack enables an attacker to acquire control over the authentication-processing requests and gain access to user data stored in Blocks. Furthermore, an attacker adds to and approves malicious transactions in the blocks, corrupting the data required by the Security Vendor to maintain records of cyberattacks on the networks and which IoT devices are infected with bot scripts.

As illustrated in Figure 7, we observe the attack resistance of the PoA consensus model in comparison with related studies. We observe the PoA has a steady transaction speed despite the growing number of block nodes being compromised due to the blocks being pre-validated. The generation and approval of blocks are granted to only a single validator, the security vendor. In the event that the number of compromised nodes grows, the validator excludes them from the list of approved blocks allowing the network to process transactions at a steady average rate of 21 transactions per second. Other proposed models. such as those proposed by Xu et al. [29], Sun et al. [28], and Lekssays et al. [27], have high resistance to DDoS attacks; however, they are compromised when the count of nodes exceeds the count of valid nodes.

### 4.4. Botnet Detection

Our botnet-detection LSTM model is based on the Accuracy, Precision, Recall, and F1 scores. Accuracy refers to packets correctly identified as benign or attack traffic and is based on the following equation:(6)$$\frac{True\;Positive+True\;Negative}{True\;Positive+True\;Negative+False\;Positive+False\;Negative}$$

Precision represents the model’s ability to correctly label benign traffic as True Negative and is represented by the following equation:(7)$$\frac{True\;Positive}{True\;Positive+False\;Positive}$$

Recall is the model’s ability to correctly identify positives as True Positives and avoid incorrectly labelling them as Negative.
(8)$$\frac{True\;Positive}{True\;Positive+False\;Negative}$$

The F1 score is represented as the mean between the precision and the recall values and is represented by the following equation:(9)$$2\times \frac{precision*recall}{precision+recall}\;$$

Figure 8 illustrates the performance comparison of the LSTM compared with existing studies. The LSTM model performs more accurately, which is attributed to the feature-selection process focusing on the mean packet arrival rate and the half-open connections. The accuracy of the LSTM is measured the highest at 99.97% after the completion of the 100th round of training. The CNN model presented by Vinayakumar et al. [25] performed the lowest in terms of accuracy, demonstrating the performance impact of converting features into a visualized format for anomaly detection. Popoola et al. [22] proposed a Federated Learning model approach, and, for comparison, we measured the final global model’s performance. Each model used a different dataset, highlighting the impact of feature selection on the model’s performance. Table 6 presents the Precision, Recall, and F1 score comparison for quantitative analysis with existing studies.

In our proposed LSTM model, the dataset is classified into test and training datasets. We implemented class weights for the two classes, benign and attack, during the training of our model. Class imbalance is determined by the assigned weights, and, in our model, weights were allocated based on the number of instances. High weightage is allocated to minority classes, i.e., those with fewer instances. Accuracy obtained the highest by 100th round, and further rounds of training did not increase the accuracy any further. The dropout value is retained at 0.2, and the number of hidden nodes increases from 23 to 64 and then to 128. The model implements a single input layer, two hidden layers, and a single output layer. Adding any further hidden layer did not improve the result. We select the output function as a sigmoid and tanh as the activation function. The LSTM model bases the accuracy of DDoS attacks on the count of packets with unique IP addresses. The accuracy, precision, recall, and F1 scores are determined using Equations (6)–(9). The study in [22] implements the Federated Deep Learning model, where the accuracy, precision, recall, and F1 score are analyzed using the similar BoT–IoT dataset as used in our LSTM botnet-detection model. However, the poor class imbalance in their training dataset of the deep neural network model results in lower performance of the precision, recall, and the F1 score. The study in [23] trains the Resnet-18 model based on several datasets; however, for fair comparison in our evaluation, we consider the results for the BoT–IoT dataset. The performance of their model is acquired from their research results. The dataset is divided into a similar 80:20 split. Finally, the study in [25] focuses on collecting DNS data using the Amrita dataset. The DNS-based approach, though, differs from the proposed LSTM model and other related work, it signifies that the impact of packet inspection yields better results. The DNS-based approach resulted in the lowest accuracy, precision, recall, and F1 scores compared to the proposed framework and related studies using the Bot–IoT dataset.

## 5. Discussion

In this section, we analyze the security of the Proposed Framework based on the five key areas of consideration for a reliable and robust framework for Digital-Twin-enabled IIoT environment from cyberattacks. Several existing studies either partially do not or entirely do not satisfy all five key areas of consideration for a secure IIoT framework from cyberattacks. Botnet detection in [22,23,24,25] relies on AI for securing IIoT networks from cyberattacks. Popoola et al. [22] implemented a Federated-Learning-based approach that guarantees Data security and Privacy by training models locally on the IoT-Edge devices. The lack of raw device data transmission to a centralized server removes the risk of man-in-the-middle-based attacks. Furthermore, the privacy of data is safeguarded as only local model updates are shared with external nodes, reducing the risk of unauthorized access to industrial data. However, Data integrity and Device Availability are chiefly unaddressed concerns, as IoT devices are of high risk to botnet attacks resulting in Poisoning attacks. Local uploaded models are trained using inaccurate data, reducing the effectiveness and reliability of the Federated-Learning-based botnet-detection method. Hussain et al. [23] implemented a dual machine learning approach, where the first model, ResNet-18, scans devices for botnet activity and the second, ResNet-19, model analyzes network traffic for DDoS attacks. The method does not meet any of the key areas of consideration, as data is flowing from IoT devices to centralized servers from model training and testing, and there are no proposed measures for identifying Delay and Replay attacks. Data security, Integrity, and Privacy are easily compromised. Device Availability is not ensured, as their proposed method relies first on identifying a DDoS attack first, and then scans devices for botnet behavior. C&C servers compromise devices and control device availability to the network, i.e., an attacker shuts down the device and wipes all sensor data. Trajanovski et al. [24] proposed an automated framework for botnet identification relying on a sandboxed Honeypot environment for trapping cyber attackers and blacklisting compromised devices. The framework addresses the identification of compromised devices after an attack takes place and thus does not ensure data security, integrity, privacy of sensor data, or the availability of devices after an attack is detected. The reactive approach in place of a proactive approach reduces its ability to secure IIoT environments from cyber-attacks. VinayaKumar et al. [25] presented a botnet-detection framework using Deep Learning to inspect Domain Name System logs for advanced persistent botnets. The deep learning model is exposed to man-in-the-middle attacks, resulting in data modification; therefore, Data security and integrity are compromised. Furthermore, the availability of devices is not addressed in the framework, as compromised devices continue to transmit data to the deep learning model and the Smart City servers.

Blockchain-based related studies [26,27,28,29] partially address Data Integrity concerns by registering all authorized devices with the decentralized network. The vulnerability of compromised new devices registered to the network reduces their ability to ensure Data security. However, each of these proposed models addresses non-repudiation, as each device is authorized and is traceable when anomalous traffic is detected. Hayat et al. [26] presented a botnet prevention method, where compromised devices are removed from the list of registered devices. The presented architecture relies on a Cloud-based model for data storage that introduces delays in the response time to attacks. Furthermore, the exposure to a single-point-of-failure vulnerability in Cloud data storage affects Data Privacy and Security concerns. Lekssays et al. [27] proposed a botnet detector for early botnet detection before an attack is initiated. The study focuses on maintaining users’ privacy using pseudo identities when initiating scans for malicious activity. However, the study does not discuss IoT device security, resulting in unaddressed device-availability concerns. Sun et al. [28] presented a Blockchain-based anomaly-detection system using device identity and attribute-based detection systems. The study focuses on preventing DoS attacks on the Blockchain network due to repeated access requests. A permissioned blockchain provides access control for communication requests; however, a lack of security measures exposes registered IoT devices to being infected with malicious scripts. The proposed system fails to prevent attackers from launching DoS attacks using registered devices. Xu et al. [29] proposed a bidirectional-linked Blockchain using a Committee Members Auction consensus algorithm for blockchain security. The vulnerability of the consensus algorithm in securing nodes is when an attacker initiates DDoS attacks on 51% of all available nodes. The studies in [27,28,29] have robust inter-device secure communication protocols using Blockchain networks for authentication. However, none of the studies address device availability concerns to prevent an infected device from installing malicious scripts on other devices.

Next-generation IIoT environments such as Smart Factories are increasingly adopting Digital Twin-based networks for the virtualized improvement of the conceptual design of products, their development, and the overall production process and reducing the overall costs of final physical implementations. The security of these DTs from cyber-attacks resulting from Botnet formation is an existing challenge. The proposed Blockchain-enabled Secure Digital Twin Framework focuses on securing the data-collection process from unauthorized entities for a reliable and robust botnet-detection model. The security of DTs is addressed to prevent incorrect and expensive conceptualization of the virtual model implemented in the physical production process, incurring a high financial burden on the factory to redesign and re-implement the DT.

In the proposed framework, each DT and the PA is registered on the Blockchain network to ensure data collection is achieved from an authorized set of IoT devices. Each authorized DT validates itself to the decentralized network before transmitting data to the Deep Learning model for botnet detection. Each data point transmitted is further verified by synchronizing the packet data collected by both DT and the PA, reducing the risk of man-in-the-middle attacks and ensuring data security and Integrity. The collection of packet data by the Deep Learning model includes only Packet Headers enabling analysis of encrypted data without requiring them to be decrypted. Deep Packet Inspection of encrypted packets is not required, ensuring Data Privacy is maintained. The detection of an infected DT is prevented from inter-digital communication by revocation of the certificate. Devices that are part of an infected digital twin are blocked from transmitting data to other machines to prevent the spread of the botnet script and ensure the availability of non-compromised IoT devices. Lastly, recording IP addresses in the Blockchain network allows the network to accurately establish and label all infected devices that are part of a DT as compromised and requires a firmware update procedure to remove the botnet script and close software security vulnerabilities.

Smart City services, such as Healthcare, Infrastructure management, Retail, and Energy management deploy DT to analyze, predict, and optimize the performance of the physical-world environment. Industries focus on adopting a proactive approach using real-time monitoring for future uptime and downtime to ensure minimal loss to quality of service. Cyberattacks are a constant concern for the Smart City infrastructure, where user-data integrity and privacy are chief concerns. The vulnerability of sensors to botnet attacks is ever growing. The proposed Blockchain-enabled Secure Digital Twin Framework allows industries to adopt a proactive approach and detect weakness in the network early, before a Man-in-the-Middle or DDoS attack takes place. Using the example of a Hospital in a Smart Healthcare environment, the design of the DTs of different patients, where each patient is connected using various sensors, enables the prediction of patient health, such as drug reaction. Sensors transmit data to their respective DT for data analysis using AI models. Man-in-the-middle attacks either prevent packet arrival at the edge layer or intercept and transmit corrupt data to poison the AI models. As devices are pre-registered in the Blockchain network, an attacker delays packet-data transmission or injects malicious data spoofing as the registered device. A separate PA collects patient data on a defined period set by the Healthcare IT department or the selected security vendor and compares the packets collected by the Digital Twins. The continuous monitoring of packets enables a Hospital to ensure that all AI models are trained using accurate patient data. If device data is altered in the transmission between the physical device and the DT, the IT manager or security vendor is alerted to a possible attack and logs the malicious Digital Twin along with its associated IP addresses of infected IoT devices in the Blockchain network. A second attack parameter involves a botnet formation where an attacker attempts to flood packets to the Digital Twin and cause a Denial of Service at the Edge layer. Using PA, the framework prevents poisoned data from being used to train the botnet model. As patient data privacy is a chief concern, only packet headers are monitored to prevent the need for decrypting packets. Infected Digital Twins are isolated from communicating with other Digital Twins and prevent the spread of botnet growth. The IT administrator or Security Vendor assigns a security firmware update to secure the physical devices from future botnet attacks. The proposed framework is especially important for future Smart City services, as the next IR 4.0 is shifting towards DT for optimizing their services. A proactive Blockchain- and DT-enabled cybersecurity framework supports future networks with a proactive approach to identify both man-in-the-middle and botnet attacks and take steps for network security before other devices are compromised.

## 6. Conclusions

This paper presented a Blockchain-enabled Digital Twin Framework for securing IIoT environments against Cyberattacks. Digital Twins of the Smart Factory are designed on the edge layer; they synchronize data with their respective IoT devices. A Smart Contract is designed to secure the registration and authentication of all network entities with the private blockchain network. Digital Twins and Packet Auditor securely authenticate and validate with the Blockchain network and synchronize data, ensuring packet data are not modified in transmission. The process flow of each process was discussed in detail. The Deep Learning-based model inspects traffic of both encrypted and unencrypted packets and identifies devices infected with malware. The device isolation policy prevents the infection of other Digital Twins, preventing the growth of botnet activity. Future work will focus on designing IP-tracing methods to accurately identify command and control servers connecting with devices using a multi-hop method. In our future work, we aim to study the impact of Mobile Digital Twins for Logistics and general Transportation services using a Cloud–Edge architecture with support for extended mobility.

## Figures and Tables

**Figure 1 sensors-22-06133-f001:**
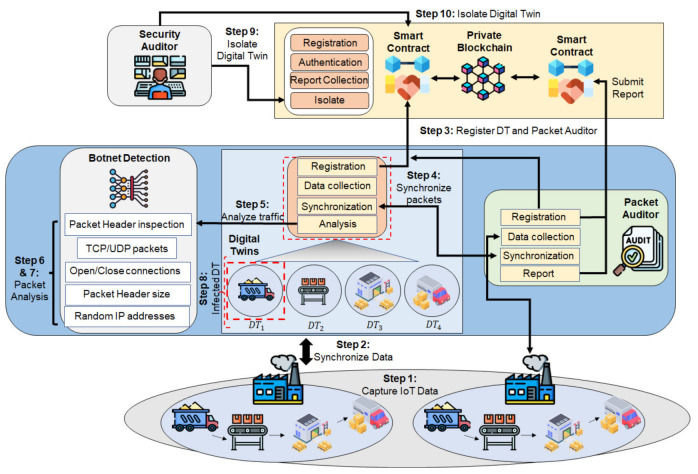
Blockchain-enabled Secure Digital Twin framework overview.

**Figure 2 sensors-22-06133-f002:**
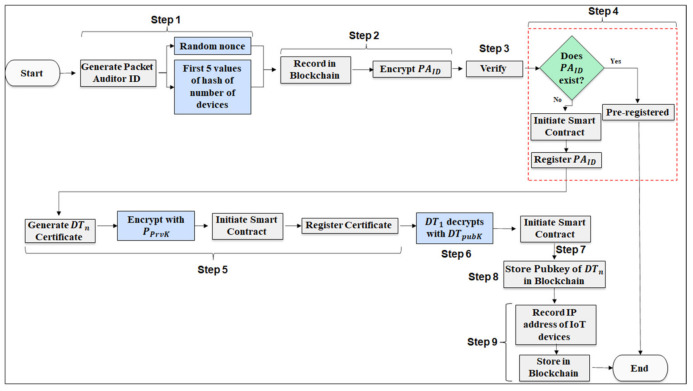
Packet Auditor and Digital Twin registration.

**Figure 3 sensors-22-06133-f003:**
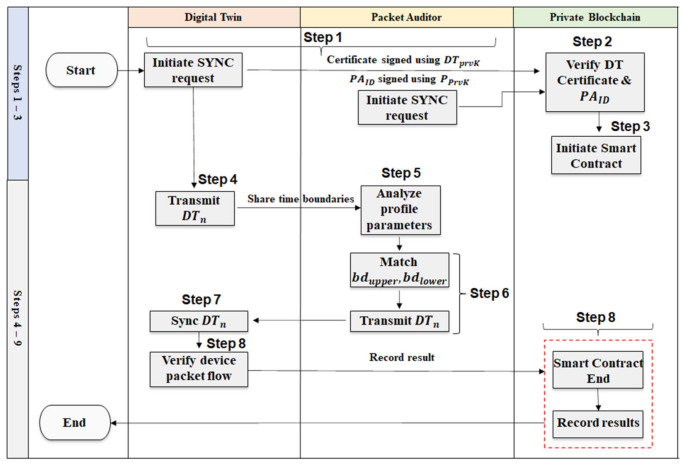
Digital Twin synchronization.

**Figure 4 sensors-22-06133-f004:**
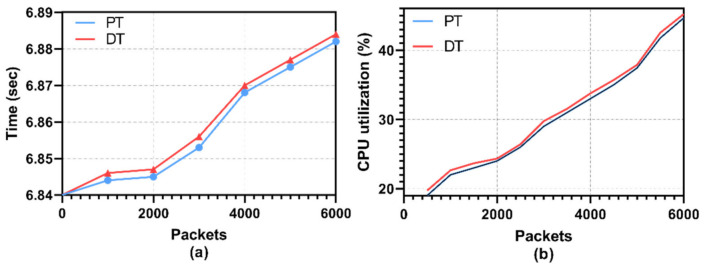
(**a**) DT Synchronization with PT; (**b**) CPU utilization.

**Figure 5 sensors-22-06133-f005:**
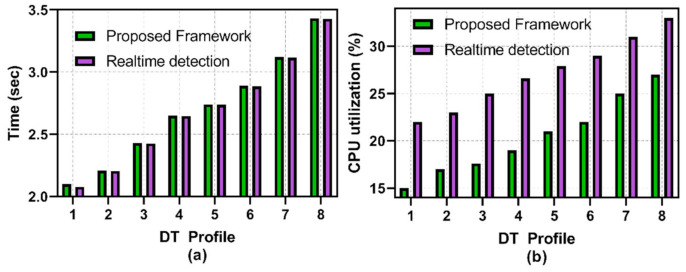
(**a**) PA packet analysis; (**b**) CPU utilization of PA.

**Figure 6 sensors-22-06133-f006:**
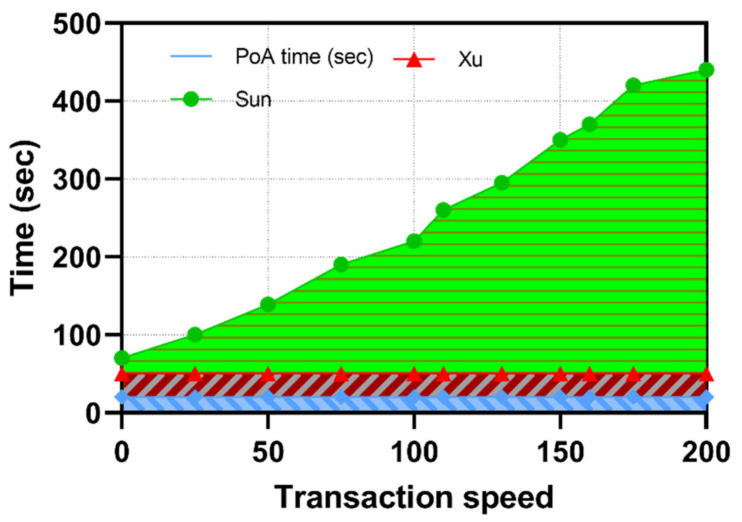
Comparison with existing studies based on block transaction speed.

**Figure 7 sensors-22-06133-f007:**
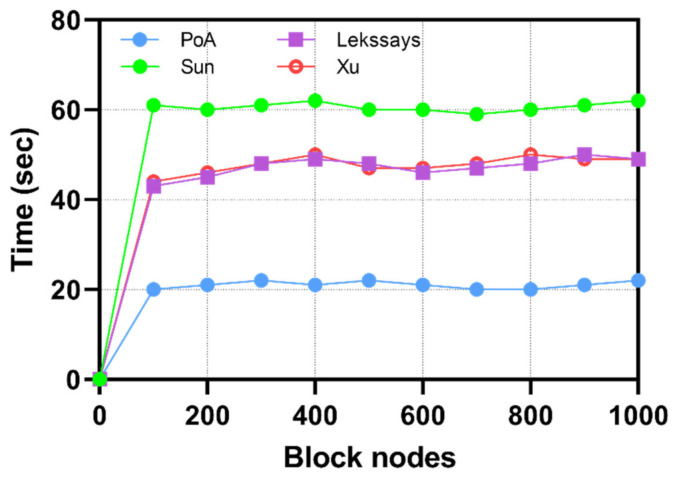
Comparison analysis of consensus algorithms to DDoS attacks.

**Figure 8 sensors-22-06133-f008:**
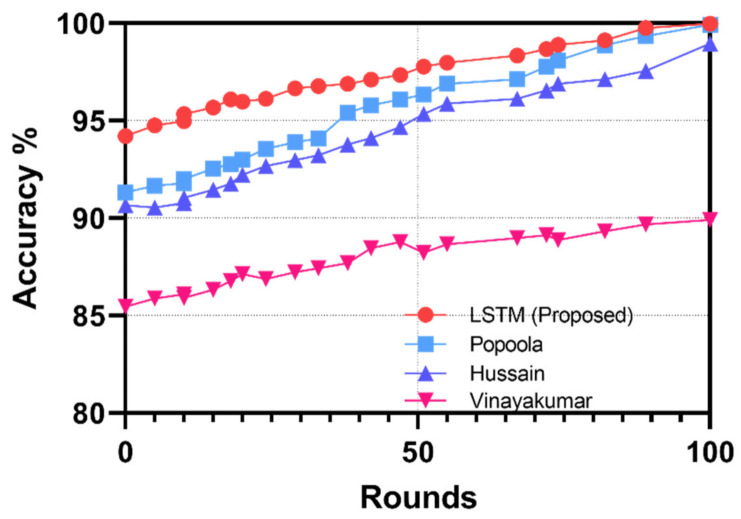
Comparison analysis of accuracy in botnet detection with Popoola.

**Table 1 sensors-22-06133-t001:** Comparative Analysis of the proposed scheme with related research.

References	Mechanism	Data Security	Data Integrity	Data Privacy	Availability	Non-Repudiation
Popoola et al. [22] (2021)	Federated Learning	Model is trained locally on devices	Poison attacks affect data integrity	Only local model gradients trained at the device are shared with the network	Device availability is not addressed in this study	Non-repudiation is not addressed in this study
Hussain et al. [23] (2021)	Dual Machine Learning	Data transmitted to centralized server is exposed to man-in-the-middle attacks	Machine Learning models train using compromised data	Data in transmission is exposed to man-in-the-middle attacks	Arithmetic operations are performed over an untrusted cloud server exposing computation process	Records of infected device are not maintained
Trajanovski et al. [24] (2021)	Honeypot	Delayed identification of compromised devices does not address data security	Delayed identification of compromised devices does not address data integrity	Delayed identification of compromised devices does not address data privacy	The research does not address device availability	Records of infected device are not maintained
Vinayakumar et al. [25] (2020)	Deep Learning using DNS Query	Man-in-the-middle attacks compromise data upload for model training	Man-in-the-middle attacks transmit corrupt data in transmission	Pseudo IDs preserve the privacy of users	The research does not address device availability requirement	Records of infected device are not maintained
Hayat et al. [26] (2022)	Machine Learning and Blockchain	Data is securely stored in Blockchain	Malicious devices are preregistered in the Blockchain network, transmitting compromised data to the Machine Learning model	Privacy of users are maintained by verifying identities at both the Edge and the cloud layer using dual signatures and identifiers	Malicious devices are ejected from the network	The study does not address recording of compromised devices.
Lekssays et al. [27] (2021)	Blockchain	Data is securely stored in Blockchain	Blockchain validates devices allowed to transmit data	Privacy of data is not addressed in the study	The study does not prevent spreading of botnet script	The study does not address recording of compromised devices
Sun et al. [28] (2021)	Blockchain and Encryption	Data storage in Blockchain prevents data manipulation	Public key-based authentication prevents corrupt data upload	The study does not address Data Privacy	The study does not prevent spreading of botnet script	Device information is stored in Blockchain for traceability
Xu et al. [29] (2021)	Blockchain and Smart Contracts	Consensus algorithm ensures stored data security	Infected IoT bots transmit data for anomaly detection	Secret keys provided to authorized members access data.	The study does not prevent spreading of botnet script	Device information is stored in Blockchain for traceability
Proposed scheme	Digital Twin and Blockchain	Authorized and registered Digital Twins share data	Synchronization between the Digital Twin and Packet Auditor verifies data transmission	Inspection of Packet Headers enables inspection of encrypted IP packets	Certificate revocation of Digital Twins prevents Botnet from spreading	IP address of infected devices are stored in the Blockchain

**Table 2 sensors-22-06133-t002:** Notation table for abbreviations used in the framework.

No.	Term	Description
1.	DT	Digital Twin
2.	$DT_{1}$	Production Floor Digital Twin
3.	$DT_{2}$	Raw Material Management Digital Twin
4.	$DT_{3}$	Assembly Line Digital Twin
5.	$DT_{4}$	Packaging and Warehousing Digital Twin
6.	HTTPS	Hypertext Transfer Protocol Secure
7.	SSL	Secure Socket Layer
9.	UDP	User Datagram Protocol
10.	IP	Internet Protocol
11.	PA	Packet Auditor
12.	$PA_{ID}$	Packet Auditor ID
13.	Txn	Transaction
14.	$Cert_{DT}$	Digital Twin Certificate
15.	$P_{PubK}$	Packet Auditor Public Key
16.	$P_{PrvK}$	Packet Auditor Private Key
17.	$DT_{pubK}$	Digital Twin Public Key
18.	$DT_{prvK}$	Digital Twin Private Key
19.	$Prof_{DT}$	Digital Twin Profile
20.	$D_{ID}$	Device ID
21.	$IPsrc_{n}$	IP Packet Source
22.	$IPdst_{n}$	IP Packet Destination
23.	$tstmp_{n}$	Timestamp of captured packet
24.	$bd_{upper}$	Upper Time Boundary
25.	$bd_{lower}$	Lower Time Boundary
26.	PT	Physical Twin
27.	DL	Deep Learning
28.	ACK	Acknowledgement Packet
29.	SYN	Synchronize Packet
30.	C&C	Command and Control
31.	LSTM	Long-Short-Term Memory

**Table 3 sensors-22-06133-t003:** Analysis of DT−PT Synchronization Latency.

Parameters	Models	1000	2000	3000	4000	5000	6000
Latency (ms)	PT	6.844	6.845	6.853	6.868	6.875	6.882
	DT	6.846	6.847	6.856	6.870	6.877	6.884

**Table 4 sensors-22-06133-t004:** Analysis of DT−PT Synchronization CPU consumption.

Parameters	Time (ms)	500	1000	1500	2000	2500	3000	3500	4000	4500	5000	5500	6000
CPU Consumption (%)	PT	19.74	22.65	23.67	24.34	26.42	29.76	31.54	33.78	35.70	37.89	42.55	45.22
	DT	19.00	22.00	23.00	24.00	26.00	29.00	31.00	33.00	35.00	37.43	41.77	44.65

**Table 5 sensors-22-06133-t005:** Analysis of $Prof_{DT}$ in Latency and CPU consumption.

Parameters	Models	1	2	3	4	5	6	7	8
Latency (s)	Baseline model	2.078	2.205	2.427	2.647	2.738	2.886	3.117	3.428
	Proposed Framework	2.1	2.21	2.43	2.65	2.74	2.89	3.12	3.43
CPU Consumption (%)	Baseline model	22	23	25	26.6	27.9	29	31	33
	Proposed Framework	15	17	17.6	19	21	22	25	27

**Table 6 sensors-22-06133-t006:** Quantitative Analysis of the Botnet-detection model with existing research.

Models	Accuracy	Precision	Recall	F1 Score
Proposed model	99.97	99.32	97.54	98.11
Popoola et al. [22]	99.93	99.08	96.97	97.96
Hussain et al. [23]	98.85	98.95	98.66	98.81
Vinayakumar et al. [25]	89.90	93.94	90.5	91.9

## Data Availability

The study does not support reported data.

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
