# Peer review of "A Blockchain-Enabled Secure Digital Twin Framework for Early Botnet Detection in IIoT Environment"

_sensors, 2022, doi:10.3390/s22166133_

Round 1

Reviewer 1 Report

-        This paper presents the blockchain initiative to enable Secure Digital Twin framework for cybersecurity in IIoT Environment.

-          I found the article is interesting and the flow of how authors present their proposed method and solutions are found good. Just, in my opinion, this paper is still lacking in term of the detailed discussion on some parts.

-          Hence, please find the following comments as my suggestion for many improvements which can be made by authors to increase the quality of your research work:

-          ABSTRACT – Need to put (IIoT) after Industrial Internet of Things. This is important as it first time appear. It becomes …. Industrial Internet of Things (IIoT)

-          LINE38 - Suggestion for improvement “The evolution of the industry (industrial revolution (IR)) from IR3.0 to IR4.0….”

-          Artificial Intelligence is used few times. This term is commonly used AI. Properly define this term and it short form and author may use it throughout your paper. I find it much better.

-          Be careful with the Capital Letter and short letter – use it wisely and where it appropriate. Cannot simply used it if it is common. For example, LINE 66 “The Framework implements a Private Blockchain managed by…” -> Framework should be framework, Private Blockchain there is suppose just private blockchain.

-          LINE 215, I found “Digital Twin (DT) of each entity…”, but the DT has been used many times in previous text. Remember, we need to declare it when IT FIRST TIME APPEAR in paper.

-          LINE 185 – The title should reflect the name of main contribution of your framework. Proposed framework of what? Too general. Please find a catchy word to show the important of your tangible result (framework).

-          Same, in Figure 1, just a proposed framework as a name of the title. Doesn’t reflect the main contribution. Give credit to your invention authors ?.

-          LINE 349, LINE 358, LINE 390: are the formula. Need to properly define in as a formula.

o   e.g: X+ Y – Z     (2.1)

-        I found few times reference to multiple source references are used, e.g. LINE 502: “Blockchain-based related studies [26-29] partially…”. Authors maybe need to try explicitly discussed and mentioned specific discussion/contribution for each reference.

Reviewer 2 Report

The paper presents a security model of digital twin. The security model is based on blockchain using deep leaning model.  Collected data from the digital twin that is connected to physical object is synchronized between the twin and its  packet header of the external IP address. Data privacy is maintained by inspecting the packet headers using packet auditor, thereby not requiring decryption of the encrypted data.  

- Lines14-15:  The authors mentioned in that " A Digital is designed for a group of devices on the edge layer to collect dice data and inspect packet hearers.."  on The other hand, in lines 215-216, the authors proposed "digital twin of each entity is designed on the edge layer..." .

Is it one digital twin for group of entity or each one has it is onw DT? Please clarify.

-  The introduction and related work should be combined to have sound literature review.  Table 1 should me moved to the the related work section and of course without the outcomes from the proposed mothed.

- Table 1 showed qualitative comparison between the existing work and the proposed work.  It will be a great added value to have quantitative comparison as well to validate the proposed model.  

- Would the model works for fix and mobile digital twins?

- More elaboration on the applications of proposed   model on  other smart space applications is a plus for the benefit of the readers. 

Reviewer 3 Report

The paper presents block chain digital framework. What were the criteria for the selection of IoT devices? It would have been better if the steps could be discussed in the form of a high-level algorithm. 

Many abbreviations are repeated and require explanation like digital twin synchronization. This paper has a low contribution in terms of scientific addition.  

Also, related work requires a detailed comparison of other published literature with the authors' contributions.

The proposed framework is not well defined. I advise authors to work on the proposed framework and list all steps in algorithm format and resubmit. Also, the entire paper is based on the proposal but the main results of the proposed framework are missing.  

Round 2

Reviewer 1 Report

I found authors have updated and taking care all comments as mentioned in my previous cycle of revision. Thank you.

Author Response

Dear Reviewer,

Reviewer 3 Report

Still, related work requires a detailed comparison of other published literature with the authors' contributions. Comparison requires key contribution comparisons with different published works. 

Kindly discuss the selection criteria for the Accuracy Precision Recall F1 score of each table. 

Author Response

Dear Reviewer,
